# Design and Evaluation of an Alternative Control for a Quad-Rotor Drone Using Hand-Gesture Recognition

**DOI:** 10.3390/s23125462

**Published:** 2023-06-09

**Authors:** Siavash Khaksar, Luke Checker, Bita Borazjan, Iain Murray

**Affiliations:** School of Electrical Engineering, Computing and Mathematical Sciences, Curtin University, Bentley, WA 6102, Australia; ltc@checker4.org (L.C.); bita.borazjani@curtin.edu.au (B.B.); i.murray@curtin.edu.au (I.M.)

**Keywords:** alternative control, finger tracking, human computer interface (HCI), hand gesture recognition (HGR), media pipe hands (MPH)

## Abstract

Gesture recognition is a mechanism by which a system recognizes an expressive and purposeful action made by a user’s body. Hand-gesture recognition (HGR) is a staple piece of gesture-recognition literature and has been keenly researched over the past 40 years. Over this time, HGR solutions have varied in medium, method, and application. Modern developments in the areas of machine perception have seen the rise of single-camera, skeletal model, hand-gesture identification algorithms, such as media pipe hands (MPH). This paper evaluates the applicability of these modern HGR algorithms within the context of alternative control. Specifically, this is achieved through the development of an HGR-based alternative-control system capable of controlling of a quad-rotor drone. The technical importance of this paper stems from the results produced during the novel and clinically sound evaluation of MPH, alongside the investigatory framework used to develop the final HGR algorithm. The evaluation of MPH highlighted the Z-axis instability of its modelling system which reduced the landmark accuracy of its output from 86.7% to 41.5%. The selection of an appropriate classifier complimented the computationally lightweight nature of MPH whilst compensating for its instability, achieving a classification accuracy of 96.25% for eight single-hand static gestures. The success of the developed HGR algorithm ensured that the proposed alternative-control system could facilitate intuitive, computationally inexpensive, and repeatable drone control without requiring specialised equipment.

## 1. Introduction

### 1.1. Background

Alternative-control algorithms consist of two main components, a non-standard human–computer interface (HCI) and a command mapping algorithm [1,2,3,4]. An alternative-control algorithm is considered successful in its application if the alternative HCI extends upon the functionality offered by the conventional control medium. Within the literature, the degree of this success is commonly appraised against the following criteria: higher accuracy, ease of use without holding any equipment or instruments in hand, shorter user learning cycle, lower cost, offers capabilities that are not available in traditional interfaces, and computationally inexpensive [5].

This paper proposes the use of hand-gesture recognition (HGR) as an alternate HCI. Gesture recognition is the mechanism by which a predefined physical action made by a user is recognized by a system [6]. HGR has been an extensively researched topic over the past 40 years [5], resulting in a plethora of different viable approaches. Modern HGR applications use a machine-learning pipeline to achieve this recognition [5,7]. Within literature, this pipeline is defined to consist of four subcomponents: data-acquisition medium, gesture description, gesture-identification algorithm, and gesture-classification algorithm [7,8,9].

The application of HGR in alternative-control algorithms for drones has been a popular area of research for several years. Various studies have investigated different combinations of HGR subcomponents to optimise the process of recognising hand gestures and translating these gestures into drone actions. In [10], the authors employed sensor fusion between a mechanomyography band and a hand-mounted initial measurement unit (IMU) to achieve robust control of an aerial drone using only the mechanical motion of the hand. In [11], the authors utilised a single RGB camera with marker gloves to recognise static gestures in combination with a hand-mounted IMU to recognise the dynamic motion of these gestures. In [12], the authors used a single RGB camera input paired with MPH and a long short-term memory neural network to achieve intuitive drone control that required no calibration or specialised equipment. In [13], the authors created a drone control simulation using a stereo camera (leap motion controller) as the primary input. In [14], the authors constructed a novel glove-based HGR system that also provided vibrotactile feedback to the system operator. The modern approaches cited above represent just a small subset of the HGR implementations that have been applied in the context of alternative control. These approaches vary in the selection of all four sub-components, the general taxonomy of known approaches for each HGR subcomponent is explained in greater detail under Section 1.2.

### 1.2. Existing Methods

#### 1.2.1. Data-Acquisition Medium

The data-acquisition sources utilized by HGR algorithms can be defined into two governing categories, these being image-based approaches and non-image-based approaches [8]. The image-based category contains the following subcategories: marker, depth camera, stereo camera, and single camera. The non-image-based category contains the following subcategories: glove, band, and non-wearable. Non-wearable technologies have been omitted from this investigation as they are an emergent technology with limited implementations available [8,15]. Marker-based approaches have also been omitted from this investigation as they have been made largely obsolete by advancements in machine perception [5,8]. The remaining viable components are illustrated in Figure 1.

#### 1.2.2. Gesture Description

There are three aspects that form the gesture descriptor within existing HGR algorithms. These are the physiological scope of the gesture, the information interpreted from the gesture, and the model used to represent the gesture [16]. All three of these factors vary greatly between the HGR implementations detailed within the reviewed supporting literature [6,16,17].

Physiological scope refers to the pre-set taxonomy used to define the physical nature of the gestures [16]. The main distinctions that are made in the existing literature when defining this taxonomy are: the use of static or dynamic gesture set, the inclusion of wrist motion, and the number of hands that are used to form single gestures.

The information interpreted from gestures by HGR algorithms has three categories: spatial information, pathic information, and symbolic information [6]. Spatial information refers to the position of the gestures within the environment. Pointing gestures are an example of gestures that convey spatial information. Pathic or temporal information is interpreted from the velocity and path that an observed gesture takes within an environment. Much like spatial information, this is typically observed from the world coordinates of the observed gesture. Symbolic information refers to the shape the observed gesture makes and is typically interpreted through joint angles’ calculation or shape-matching techniques [6]. 

Within HGR algorithms, the model used to represent an observed hand changes to reflect the desired scope of input gestures [16]. As the number, complexity, and information density of gestures increase, the complexity of the modelling method used must also increase. Model complexity is directly proportional to the number of classifiable landmarks that the model provides [16]. HGR modelling methods fall into two main categories: 3D-based models and appearance-based models. These models are demonstrated in Figure 2.

Less complex models such as the silhouette geometry model are preferably used for simple HGR applications as they offer very few classifiable landmarks. Given that it is an appearance-based model, it is computationally inexpensive to generate as it can be extracted directly from the image with little intermittent computation. These model styles are best suited for low-response-time algorithms that specialize in lightweight and fast operating applications. More complex models such as the 3D-skeleton model typically offer up to 21 landmarks [16,18,19] for classification. These models require considerably more computational power to generate accurately, but the 21 landmarks enable the calculation of exponentially more distinguishable gestures. Due to the higher computational load required to generate the model, they are typically employed in control applications where a higher accuracy and a more expansive data set are required [20].

#### 1.2.3. Gesture Identifiers

Gesture identification is a catch all term that refers to the method by which a human hand is detected as apart from its background and transformed into a computer model used for classification [5,8]. This process is often referred to as feature extraction [21]. The model referenced is an estimation of the human hand, and the model type is as described in Section 1.2.2. The observational method used to collect the data from which the hand is detected is the data-acquisition component discussed in Section 1.2.1. As there are a multitude of different approaches for each combination of model and observation method, this review focuses on modern methods that utilize single-camera visual observation methods and 3D-skeleton representations [22]. 

Media pipe hands (MPH) is a complete and well researched on-device real time hand identification solution designed to operate using a single RGB camera [18]. The output produced is a list of 2.5D, 21-landmark skeleton models for each hand observed within the input frame. MPH utilizes a computationally efficient two-stage pipeline: the first stage is a palm detector, and the second stage is the hand-landmark extraction method. This pipeline was designed to minimize the computational load of 3D-skeleton identification in two key methods. The first method uses a computationally inexpensive algorithm to locate areas of interest within the image and then applies the landmark model only to these areas. The second method uses the tracking of identified hands between frames to reduce the computation requirements necessary to perform identification of the subsequent frames.

InterHand2.6M (IHM) is a relatively new gesture-identification algorithm that utilizes a single RGB camera and a pre-trained convolutional neural network (CNN) labelled ResNet to achieve highly accurate feature extraction [19]. The output produced is a normalized 3D, 21-landmark skeleton model for up to two hands, specifically tuned to detect and correctly label the left and right hands of a single operator [19].

#### 1.2.4. Gesture Classifiers

Gesture classification refers to the process by which a feature extracted by the gesture-identification algorithm is classified as a particular gesture from a pre-defined list [5,21]. The classification of input-gesture models is a typical machine-learning problem and can be addressed by numerous different algorithms. Popular approaches include decision trees, K-nearest neighbours (KNN), the hidden Markov model (HMM), artificial neural networks (ANNs), naïve Bayes (NB), linear regression, bounds-based classification, support vector machines (SVMs), and convolutional neural networks (CNNs). Modern HGR approaches favour the use of classifiers that can handle high-dimensionality features spaces and classify elements into many distinct non-linearly separable classes. 

### 1.3. Contribution of the Paper

The primary contribution of this paper is the development of a cohesive, high accessibility, low-cost, alternative-control algorithm. This paper used a multi-stage method to identify, analyse, and clinically validate modern HGR components such as MPH. A cohesive alternative-control algorithm was constructed by using the results of these analytical stages to implement components that best complemented one another and satisfied the overarching design criteria of the paper. This set of complimentary HGR components operated at a high level of confidence and robustness against gesture confusion. The developed HGR algorithm had a gesture-classification accuracy of 96.25% over an array of eight input gestures, which is comparable to modern HGR algorithms [10,11,12,13,14]. The final alternative-control algorithm was demonstrated using a quad-rotor drone, whereby the algorithm was able to address its core developmental criteria and extend the functionality of the drone’s conventional control medium. In comparison to modern alternative-control systems for drones, the final algorithm presented in this paper provides an increased level of accessibility, a higher computational efficiency, and a lower monetary cost. The increased level of accessibility was achieved due to the clinical validation of MPH, which led to the development of HGR systems that compensate for the Z-axis instability of the MPH model. This meant that users are no longer required to maintain ideal hand orientation in front of the input RGB camera, making the system easier to operate than other approaches that utilise MPH [12]. Furthermore, the final solution does not require specialised equipment, further increasing its accessibility and lowering its monetary cost [10,11,13,14]. Finally, the computational requirements of the final solution are minimal when compared to other vision-based HGR alternative-control algorithms [12,13].

The results achieved in this paper support the use of HGR algorithms such as MPH in alternative-control applications. Thus, the framework used to develop the alternative-control algorithm detailed in this paper could be re-applied to a multitude of other applications by simply reconfiguring the command mapping component. The secondary contribution of this paper is the novel validation of MPH modelling accuracy. These data can be used to inform future projects on how best to apply MPH to their respective applications even in project scopes that extend beyond alternative control. 

## 2. Methods

### 2.1. Methodology Structure

#### 2.1.1. Overview

The investigatory scope of this paper focused on using pre-existing gesture-recognition components to construct an alternative-control algorithm. This methodology section is focused on the selection, validation, and implementation of these standalone components to form a cohesive final algorithm. To achieve this, each standalone component was investigated, with subsequent investigations being adjusted to reflect the results of the previous stages. This overview describes the structure of the investigatory method, the governing criteria used to define each of the investigations, and the two key simplifications used to manage the scope of the overall investigation. 

Alternative-control algorithms consist of an HCI component and a command mapping component. The HCI component selected for this project was HGR. From the literature reviewed in Section 1.1, all HGR algorithms consist of the following four components: gesture description, data-acquisition method, gesture identifier, and gesture classifier. With the inclusion of a command mapping component, the list of components required by the final solution was derived. The subsequent investigation used to select a final implementation for each of these components was broken up into six stages. The first stage was the selection of a gesture-description model. The second stage was the selection of a data-acquisition method. The third stage was the selection of a gesture-identification algorithm. The fourth stage was the validation of the selected gesture-identification algorithm. The fifth stage was the selection of a gesture-classification algorithm. The sixth and final stage was the derivation and tuning of a gesture-mapping component. The general structure of the investigation is shown in Figure 3. 

#### 2.1.2. Defining Simplifications

As demonstrated in Section 1.2, due to the mature nature of HGR, there were numerous unique approaches available for each of the four algorithm subcomponents. Given the expansive scope of viable approaches, it was not feasible to investigate all possible subcomponent combinations directly. Consequently, to maintain the validity of the investigation, three simplifications were defined to manage the scope of the investigation: Components selected prior to the current stage of the investigation cannot be changed. Implementations were only considered if they were applicable with the previously selected components. Selected components should not be changed to accommodate for the needs of a new proposed approach. For example, the data-acquisition method selected in stage two could not be changed to accommodate for the requirements of a gesture-identification component proposed in stage three.The selection of each component was to be made without consideration for future components. The selection of each component was to be based on the applicable governing criteria, selection of previous components, and the relevant validation results.The gesture-description and data-acquisition components were selected from the reviewed literature, without any new quantitative or qualitative analysis being performed.

Simplifications 1 and 2 enforced a linear investigation structure. By using this linear method of selection, the number of applicable implementations reduced exponentially with each subsequent stage of the investigation. This reduced the scope of each investigation stage to a workable level whilst maintaining the validity of the overall investigation. However, these assumptions only worked to reduce the scope of the later stages of the investigation. Hence, the inclusion of a third simplification was required to reduce the scope of the earlier stages. 

The third simplification removed the need for time-consuming experimental analysis to be performed during the first two investigatory stages. This reduced the workload required to analyse different implementations, allowing for a wider array of implementations to be investigated. This simplification did not compromise the validity of the overall investigation for two key reasons. The first reason is that it is difficult to qualitatively or quantitively analyse the effectiveness of different gesture-description and data-acquisition methods without considering the HGR algorithms that are typically included within. The most effective way to analyse complete solutions is to review the literature used to define them, hence supporting the selection of these components following a literature review. The second key reason is that the criteria relative to these two components are not defined by analysable metrics, and as such can clearly be derived from the attributed literature in a yes or no fashion. 

By utilising Simplification 3 to inform the selection of the highest-level components (Components 1 and 2), and then enforcing Simplifications 1 and 2, the scope of each individual section can be managed appropriately—facilitating an efficient derivation of the final solution. Furthermore, the application of Simplification 3 enables the selection of Components 1 and 2 within Section 2.2 and Section 2.3 respectively, as no experimental investigation was needed. This allowed for more specific testing to be defined for subsequent sections.

#### 2.1.3. Governing Criteria

To ensure that a cohesive and effective final solution was developed over the course of this investigation, a governing set of criteria were developed. The purpose of these criteria was to augment the dependent and independent variables used within each investigatory stage, in a way that led to the development of an effective alternative-control algorithm. That is, if the implementation selected for each component is selected because it conforms best to the defined criteria, then the final solution will function as an effective alternative-control system. The selected criteria were a subset of the list found in Section 1.1 of the literature review. These criteria were selected from the larger list to specifically adhere to the design approaches used in [3,4] which successfully produced enervative and effective solutions. The final set of governing criteria used in this project are listed below in order of importance:4.Reliability in issuing the intended command: This criterion graded both the number of unique commands the algorithms can issue, and the algorithm’s ability to distinguish between these unique commands.5.Reproducibility of the intended command: This criterion graded the algorithm’s ability to robustly reproduce the same action when presented with the same user input.6.Physically non-restrictive equipment or instrumentation: This criterion graded how restrictive the algorithm’s control interface was, referring to both restrictions caused by elements that were physically placed upon the user’s body or elements that required the user to operate within a restricted space or in a restricted manner.7.Ease of operation and shorter user learning cycle: This criterion graded how complex or difficult to learn the control interface was, considering factors such as complexity of inputs, complexity of input structure and physical difficulty to form inputs.8.Computationally inexpensive: This criterion graded the computation requirements for the algorithm’s operation and inversely the speed in which the algorithm could operate at if given an abundance of computational resources.9.Monetarily inexpensive: This criterion graded the general cost needed for the algorithm to function, including the cost of the computational hardware required to run the algorithm and the sensory hardware to acquire data.

### 2.2. Stage One: Selection of Gesture-Description Model

#### 2.2.1. Overview

The first stage of the investigation selected a gesture-description method that could facilitate specific and repeatable control of a complex system. This stage was affected by Simplification 3, which specified that the investigation type was a review and not a quantitative or qualitative investigation. As stated in Section 1.2.2, the gesture-description component of an HGR algorithm consists of three subcomponents: gesture type, gesture information, and gesture model. All three subcomponents were evaluated individually and selected per Governing Criteria 1, 4, and 5. 

Due to the relevant simplification, the method used to determine the final implementation for these three subcomponents was a literature review. The options to be reviewed for each of the subcomponents were derived directly from the reviewed literature and are listed under each of the subcomponent’s justification sections. 

The final values selected for these fields are as follows: for the gesture type, single-hand static gestures were chosen; for the gesture information, symbolic information was selected; for the gesture model, a three-dimensional, 21-landmark skeleton model was selected. These values were selected for two key reasons. The first was to ensure that the final solution met the relevant governing criterion, and the second was to ensure that the simplest solution to these requirements was found. 

#### 2.2.2. Gesture-Type Selection Justification

The type of gestures observed had three main considerations to be analysed. The first of these was the motion of observed gestures, either a static or dynamic gesture set. The second was the scope of observation, specifically the inclusivity of wrist motion. Finally, the scope and number of observable gestures was considered.

Static single-hand gestures were selected to ensure future components’ simplicity and ease of understanding for operators. While dynamic or two-handed gestures also conform to Criterion 4, the construction of an algorithm that operates using these gestures would have been considerably more complex than an algorithm designed to recognise static gestures [16]. Furthermore, single-hand static gestures are easier for an operator to learn and perform consistently, decreasing the user learning cycle in comparison to the more complex dynamic or two-hand gestures [16]. Thus, single-hand static gestures were selected to simplify the computational restrictions and user learning cycle of the final solution. 

A wide range of gestures was initially proposed to avoid artificially biasing the identification and classification components’ selection by providing an ample array of gestures. The only restriction on the initial gesture set was that it had to contain gestures defined by a recognised sign-language system. This was performed to ensure that the selected gestures were easily recognisable and easily learned [22], aiding in the final solution’s conformity to Criterion 4. An example gesture set is shown in Figure 4.

#### 2.2.3. Gesture Model Selection Justification

The gesture-model selection was governed by two key factors, the computational complexity required to generate each model and the number of classifiable landmarks offered by each model. This selection aimed to balance these two factors by selecting a modelling method that allowed for enough classifiable landmarks to differentiate between the types of gestures detailed in Section 2.2.2, whilst not requiring excessive computational power to generate. The modelling methods analysed are listed in Section 1.2.2. Given the number and complexity of the possible gestures, the most applicable model was a 3D-skeleton model [18,19]. Appearance-based models were not applicable as their low number of classifiable landmarks would limit the final algorithm’s ability to differentiate between the desired input gestures [16]. More complex 3D models, such as 3D geometric models and 3D textured volumetric models, were not applicable as the additional classifiable landmarks they offer are not necessary to differentiate between the desired input gestures. Thus, the use of these models would needlessly increase the complexity of future components without benefitting the final algorithm’s performance [16]. 

#### 2.2.4. Gesture Information Justification

The information derived from the gestures had four considerations, these being spatial, pathic, symbolic, or affective [6]. Note that the selection of these information sources was not mutually exclusive, i.e., one or all of them could be selected. Given the gesture type selected in Section 2.2.2, symbolic information was the primary source of information that was extracted from the observed gestures [6]. Additionally, the spatial information of the three-dimensional skeleton landmark model was also used to calculate the joint angles for each of the 15 observed joints. Stage four methodology will define the specific calculations required to perform this transformation.

### 2.3. Stage Two: Selection of Data-Acquisition Method

The second stage of the investigation was the selection of a data-acquisition method. The purpose of this stage was to select a data-acquisition method capable of efficiently and non-restrictively observing a human hand in a manner conducive to the production of the selected gesture model. Similar to Section 2.2, the investigation process for this stage was a literature review as defined by Simplification 3. The governing criteria relevant to this section were Criteria 1, 2, 3, 5, and 6. To ensure that these criteria were satisfied, this review assessed all of the HGR data-acquisition methods listed under Section 1.2.1. Each of these solutions were analysed against the criteria listed above and compared against one another to find an optimum solution. 

Out of the analysed data-acquisition methods, single RGB cameras were the only analysed approach that satisfied the applicable governing criteria. In contrast, the other data-acquisition methods all posed notable drawbacks that would severely hinder the final solution’s ability to satisfy these criteria. Specifically, depth cameras were omitted due to their range and availability restrictions, which would jeopardise the final solution’s conformity to Criterion 3 as the range restrictions will restrict users [8,24]; stereo cameras were omitted due to their extensive computational requirements and focal pointing restrictions, which would make satisfying Criteria 3 and 5 difficult [7,8,13]; and band and glove approaches were both omitted because of their direct opposition to Criterion 3 [8,10,11,14,15].

After these omissions, the single RGB camera was the only remaining viable approach. However, single-camera approaches have some notable drawbacks that will need to be addressed by future components. These are primarily the robustness issues associated with background and operator hand, variability, and single viewpoint error sources such as self-occlusion and transform inconsistency [8]. Despite these notable drawbacks, due to the mature nature of this form of HGR [7], it is reasonable to assume that the selection of appropriate future components can appropriately manage these drawbacks [7,12].

### 2.4. Stage Three: Selection of Gesture-Identification Algorithm

The purpose of stage was to select a gesture-identification algorithm capable of extracting hand features from the data returned by a single RGB camera. The extracted features were to be arranged in the form of the desired three-dimensional skeleton model. The governing criteria relevant to this section were Criteria 1, 2, and 5. Additionally, the prospective algorithms were also investigated as to their ability to minimize the drawbacks of single RGB camera approaches, such as self-occlusion. A qualitative analysis was performed to facilitate this selection, focusing on the computational cost of the implementation and the observable localization accuracy of the prospective algorithms. This stage was only intended to be a minor thresholding investigation, aimed less at comparing applicable solutions and more towards ensuring the selected solution will be able to conform to the governing criterion.

Given that there are a multitude of feature-extraction methods that are applicable for HGR gesture identification, it was not feasible to test them all directly. Fortunately, this expansive scope was reduced considerably by the three design simplifications. The previously selected gesture-description and data-acquisition components reduced the scope of this investigation in the following ways: the removal of any method not initially developed to return a three-dimensional skeleton model; the removal of any gesture-identification approach not compatible with single RGB camera data; and only considering pre-existing open-source implementations. After these reductions, three solutions were marked for future investigation: media pipe hands, InterHands2.6M, and an OpenCV approach. These three algorithms represented possible solutions that applied different pre-processing and feature-extraction techniques and had drastically different computational loads. 

The key dependent variables of this qualitative investigation were the localisation accuracy of the skeleton model and the computational requirements needed to perform feature extraction. These two dependent variables were analysed in two subtests. The first test aimed to observe the computational requirements to set up and operate the three algorithms. The second subtest aimed to observe the localisation capacity of the three algorithms in variable environments. 

10.Identifier Implementation: The aim of this subtest was to implement a baseline variant of the three algorithms. The baseline variant of this method should be capable of observing a single human hand and printing the angle of its 15 primary joints to the terminal while also displaying the 3D-skeleton model on screen. The method used to calculate these joint angles is described in Section 2.5. The purpose of this stage is three-fold. Firstly, it serves to assess the operational readiness of the algorithms. Secondly, it assesses the computational requirements necessary to implement the algorithms. Finally, it acquires an operational version of said algorithms upon which future testing would be performed. The key independent variables of this test are the three different algorithms being tested. All algorithms are to be applied on the same 2017 Mac Book Pro that operates using a 3.5 GHz Dual-Core Intel Core i7 CPU, an Intel Iris Plus Graphics 650 1536 MB graphics card, 16 GB 2133 MHz LPDDR3 of RAM, and 250.69 GB of storage.11.Qualitative Analysis: The aim of this subtest was to qualitatively observe the implemented algorithms’ localisation accuracy. This subtest was the first step towards ensuring that the selected algorithm conforms to Criteria 1 and 2 and minimizing the drawbacks of the selected data-acquisition method. This investigation stage aimed to observe each algorithm’s accuracy in cases of self-occlusion, rotation, and translation using the operational version derived in the first subtest. The method for this observation was relatively simple. First, a user’s hand was held in a constant position in front of the camera. The displayed three-dimensional model was then recorded. From this position, the hand was then rotated and translated around the camera’s viewport. While these rotations and translations occurred, the displayed model was constantly observed. These observations aimed to determine whether the algorithm could maintain its localisation accuracy despite the movement. The final stage of the observation was to turn the hand so that certain aspects of the hand become occluded from the camera’s POV. This was conducted to determine whether the algorithm could still produce a model despite the occlusion of hand features.

### 2.5. Stage Four: Validaton of Selected Gesture-Identification Algorithm

The fourth stage of the investigation focused on the validation of the selected gesture-identification algorithm. Specifically, this stage centred around the evaluation of the accuracy and robustness of the model produced by the selected algorithm. This stage employed a clinically advised, quantitative approach that compared the joint angles derivable from the model with the joint angles measured with a finger goniometer. The results of this method were then used to determine the current algorithm’s conformation to Governing Criterion 2. Due to the extensive nature of this validation process, it was not possible to apply it to each of the gesture-identification algorithms analysed in Stage 3; hence, it was only applied to validate the final selected identification algorithm.

The key dependent variable observed during this investigatory stage was the percentage accuracy of the generated model. The generated model, as specified in Stage 1, was a three-dimensional skeleton hand model. The accuracy of the model was found by calculating the percentage variance between the joint angles of the observed hand measured by a goniometer and the joint angles calculated from the generated model. Joint-angle comparison was used over other possible methods such as landmark-accuracy analysis or joint-positional analysis because of the clinical support available for joint-angle measurement. As a result of this, the joint angles could be measured directly and accurately using a clinically defined method which provided an excellent reference value for comparison with the model. In contrast to this, if landmark-accuracy analysis or joint-positional analysis methods had been applied, considerable sources of error could have been introduced into the reference value due to sources such as hand-size variation, joint-location variation, and joint-observation variation. 

The joint angles to be tested are the joint angles of the metacarpophalangeal, proximal interphalangeal, and the distal interphalangeal joints of all fingers including the thumb. For all joints’ measurements, the static arm of the finger goniometer was to be stabilized against the proximal side of the joint with the hinge of the goniometer being placed directly above the observed joint. If the participants knuckle was bulbous in nature such that it prevented the goniometer from securely sitting above the joint, then the goniometer was to be moved to the side of the finger, such that the goniometer’s hinge sat directly in front of the observed joint. Once secured, the free arm of the goniometer was then lightly pressed against the distal side of the observed joint. It is imperative that little to no force is applied during this process as the goniometer is designed to move freely, and any excessive application of force could alter the pose of the observed hand. Once the free arm of the goniometer had contacted the distal side of the joint, the joint angle could then be recorded to the nearest 5°. This method was advised by Jayden Balestra [25]. 

Two methods that were applied to calculate the joint angles from the three-dimensional model. The first method analysed was a conventional three-dimensional vector angle calculation which first created two vectors: one traveling to the desired joint from the previous attached joint and a second vector traveling from the joint to the next joint. Once these vectors had been defined, a simple dot product calculation was then applied to calculate the angle that existed between the two joints. A second method was used as a backup, which simply ignored the depth component of the model and then performed the same calculation performed above. This method was included to quantify whether the three-dimensional nature of the model was aiding or limiting the model’s performance. The code used to perform both calculations is shown in Appendix E. 

In order to thoroughly test the robustness of the algorithm alongside its accuracy, two independent variables—hand pose and hand orientation to camera—were changed throughout the course of this stage: 12.Hand pose: The first of these variables was the pose of the hand. In total, three positions were investigated: a fully closed position, a partially closed position, and a fully open position. These three positions were selected because they are stable, easy to hold, repeatable positions, and again, because they were the advised positions suggested by our clinical reference, Jayden Balestra [25]. Furthermore, these three positions were used to simulate a full range of motion of the human hand, as it was important to validate the accuracy of the model across a hand’s full range of motion. An example of the three poses used are shown in Appendix B and Appendix C.13.Hand orientation with respect to camera: The second independent variable that was altered over the course of this analysis was the incident angle formed from the camera’s point of view and the observed hand. By changing this angle of orientation, the algorithm’s robustness against rotation and self-occlusion (the drawbacks of single camera RGB solutions) could be quantitatively observed. For each of the three hand poses defined above, four photos were taken: one from directly in front of the hand, one from a 45° offset, one from a 90° offset, and one from a 180° offset. An example of the four viewpoints used are shown in Appendix B and Appendix C. To ensure a high level of accuracy within the test itself, a wide range of controls were put in place, to make sure each stage of the analysis was repeatable and accurate.14.Lighting: To avoid lighting variance, all tests were to be conducted in a well illuminated environment, specifically aiming that no shadowing be present on the observed hand.15.Background: Background variation is known to have an impact on the MPH modelling process. As this is not a factor currently being analysed, a white backdrop was used for all tests. A white backdrop was used to ensure that there was a high level of contrast between the hand and the background to aid in the feature-extraction process.16.Pose stability and body position: To ensure that the same position and viewpoint angles were observed for each participant, two controls were put in place to manage body position and hand stability. The first control is that participants are to kneel in a comfortable position, with their forearm braced against the test bench. The test bench is to contain a set of marks, indicating the appropriate positions for the background and participant forearm.17.Pitch, roll, and yaw of the camera: To ensure that the viewpoint orientation was maintained across all participants, and only varied by the desired amounts between tests, the Halide camera application was used [26].18.General hand size/distance from camera: Whilst changes in participant hand size were unavoidable, to avoid exacerbating these variations, a fixed camera distance was used for all participants. This was performed by simply having fixed mounting points for the camera on the test bench at the correct location and orientation for each of the desired viewpoint angles.

The final testing procedure consisted of five stages; (i) establish the aforementioned controls; (ii) the participant forms required hand pose; (iii) record the joint angles using goniometer; (iv) photograph the hand from the required viewpoints, (v) re-measure the joint angles using a goniometer. After the above procedure had been completed, the two sets of measured joint angles were compared. If the results of the second set of measurements failed to match the first, the test images were discarded, and the process was repeated. This was conducted to confirm that the participant’s hand pose had remained stable throughout the test. This process was repeated for each participant and for each pose. A five-minute pause was taken between tests to ensure that participant fatigue did not affect pose stability. The valid photos were passed to the selected gesture-identification and angle-calculation algorithms, which generated a set of observed joint angles (the code used to perform this stage of the process is referenced through Appendix E). These observed joint angles were then compared against the goniometer measurements to generate a final set of accuracy percentages.

### 2.6. Stage Five: Selection of Gesture-Classifcation Algorithm

The purpose of the fifth stage of the investigation was to use a quantitative method to select the gesture-classification algorithm that best complimented the selected gesture-identification algorithm. The three relevant criteria for this stage are Governing Criteria 1, 2, and 5. From these three criteria, two quantitative metrics were calculated to inform the final selection. These metrics were the classification accuracy of the tested algorithms expressed in the form of confusion matrices and classification speed expressed in seconds (used to reflect the computational requirements of the algorithms). 

To ensure that the selected classifier complimented the selected gesture-identification component, there were two possible scopes and criterion weightings defined for this stage of the investigation. Each of the defined scopes was focused on a different possible outcome which could have arisen from the results of Stage 4. Before defining the individual scopes, there was another key scope reduction that applied to both cases. The selected classifier must be a pre-trained solution, capable of classifying gestures of a globally recognised sign-language. This reduction was made to conform with the paper problem statement, and the selected gesture-description model. These two possible scopes are defined below: 19.If the Stage 4 results show that the selected gesture-identification algorithm can accurately and robustly produce a model that reflects the user’s hand, then a low dimensionality classifier built around the 15 single-dimension joint angles should be investigated. The final selected algorithm is that which favours Criterion 5 over Criteria 1 and 2. The algorithms to be investigated are decision trees, KNNs, and linear regression [27].20.If the Stage 4 results show a less than ideal model accuracy, then a higher input dimensionality classifier which uses the original 21 three-dimensional coordinate system (63 total dimensions) would be investigated. The final selected algorithm is that which favours Criteria 1 and 2 over Criterion 5. Specifically, the algorithms to be investigated are ANNs, SVM, linear regression, and a non-machine learning bounds-based approach [27,28].

Regardless of the selected scope, the experimental method that would be used to analyse the prospective classifiers remained the same. In either case, the dependent variables of the investigation remained the classification accuracy and classification time. The independent variables of this classification were the style and implementation of the classifiers themselves. To ensure a fair investigation of the defined classifiers, the following variables were kept constant: 21.Test data set: A custom data set was to be made for the selected classifiers. Ideally, after the initial group of prospective classifiers had been defined, a common set of ten gestures would be identified between the algorithms. Once this common gesture set had been defined, ten images were created for each gesture and converted into three-dimensional models using MPH. These 100 models formed the test data set for this stage of the investigation. Note, to ensure that Criteria 1 and 2 were assessed correctly, the hands present in the ten selected images varied, in scale, orientation, and pose. By introducing these variations into the common data set, the algorithm’s accuracy will be tested in a more robust fashion as they are not being tested in a ‘best-case scenario’.22.Computation power provided to each algorithm: To ensure that no one algorithm is favoured during this analysis process, all testing should be performed on the same device, with no background processes running. When performing classification-time testing, the time taken should only be considered for the ‘prediction stage’ of the classifier. Specifically, this time value should exclude the time taken to initialize/train the classifier, load the MPH model, and any time associated with the creation of the confusion matrices.

The procedure for this investigation was relatively simple, a basic algorithm was used to sequentially test each of the prospective pre-trained algorithms against the common data set. After each test, the prediction of the four classifiers was recorded in the respective confusion matrices, and the time taken to perform that classification was stored in a CSV file. Once all the test images had been fed into the algorithm, the final confusion matrices, and time performance data were displayed on screen for evaluation.

### 2.7. Stage Six: Gesture Mapping and Tuning

The purpose of this stage of the investigation was to develop a gesture-mapping component capable of translating classifiable gestures into drone actions. The mapping method selected for this component was a one-to-one command mapping approach. This approach was selected to conform with the design decisions made in [1,2,3,29] and served as a good initial solution capable of demonstrating the functionality of the fully developed HCI algorithm within the context of alternative control. This stage consists of two key developmental sections. The first section was the initial declaration of a gesture dictionary that translated observed gestures into commands. The second section was the tuning of these commands to allow for smooth control of the drone. The primary relevant criterion for this stage is Governing Criterion 4.

The DJI TELLO quad-rotor drone (mechanism specifications provided in Appendix F) was selected to be controlled by this mapping component for three key reasons. The first reason is that the physical characteristics of the drone made it well suited for use in prototype implementations such as this. The drone is inexpensive, light, and slow and has built-in collision recognition sensors and systems. These collision-mitigation sensors limited the consequences experienced during testing, which was beneficial given the experimental nature of the applied control algorithm. The second reason is that while DJI TELLO does not offer a python API, the mobile application offered by DJI to operate the drone has commands that can be easily replicated using Python’s built-in socket library. This meant limited work was required to transfer controlling commands to the drone. The final key benefit is the breadth of commands offered within the TELLO app. The TELLO app offers numerous distinct commands ranging from simple operations, such as lift-off and land, to complex tasks such as ‘do a barrel roll’.

The TELLO drone accepted two basic movement command sets, each containing six unique commands. The first command set uses positional commands that move or turn the drone by a fixed amount per transmitted command. The second command set uses velocity commands, whereby each command updates the drone’s velocity in a certain way. With the addition of the take-off and land commands, two unique sets of eight commands were defined for investigation. In either case, the eight commands were mapped to the eight gestures, most accurately classifiable by the completed HGR algorithm. Once this mapping was complete, three key factors had to be experimentally tuned through a set of test flights. These factors were: the time a gesture must be held before a command is executed, the magnitude of a response once a command is executed, and the refresh rate for command execution. The final command set and values for each of the above factors were selected because they facilitated the control that best conformed to Criterion 4.

## 3. Results

### 3.1. Gesture-Identification Selection

#### 3.1.1. Implementation Results

Three algorithms were considered and attempted to be implemented; these were media pipe hands (MPH), InterHand2.6M, and an OpenCV extension of MPH titled CVZ. However, due to the computational requirements to both train and operate, InterHands2.6M was removed from further analysis during this stage. The training data required a 365-GB download to attain all the necessary data to train the ResNet network for 30-fps operation. The 365-GB download exceeded the total storage capacity of the host machine and as such could not be completed. 

MPH was successfully implemented using its open-source solution that included APIs for both java and python [30]. The python API was used for this project. As mentioned in the Section 1.2.3, the computational requirements of MPH are minimal, as such it was expected that MPH would be able to run at a high frame rate on the selected host machine. MPH conformed to this expectation, continually maintaining its capped frame rate of 30 fps, validating its conformity to Governing Criterion 5. The output of this implementation is shown in Figure 5.

The joint-angle calculations were handled by a python module that applied the calculations described in Section 2.5. This module parsed the ‘hand’ objects produced by MPH to acquire the 3D coordinates of the joints. These 3D coordinates were converted to joint angles using the aforementioned calculations before displaying these angles using the ascii art hand shown in Figure 5. An ascii art hand was used to visually display the joint angles in the terminal, in order to ensure that they could be easily interpreted by the user. 

CVZ [31] was successfully implemented using its python API. Despite its lack of supporting literature, CVZ was relatively easily to implement. CVZ only required one variable to be set which was the webcam input source directory. Once operational, CVZ’s performance appeared to fluctuate greatly with frame rates ranging from 16 fps to 30 fps. Unfortunately, CVZ also suffered from some considerable stability issues and had a re-occurring bug that would crash the code repetitively whenever two hands appeared in-frame together. The initial output of CVZ is shown in Figure 6.

As CVZ had the same 21-landmark model structure as MPH, only minor changes were needed to be adapt MPH’s joint-angle calculation python module to match CVZ. The only changes required were to adjust the code pair with CVZ’s landmark data structure. After these changes, the joint information was able to be successfully calculated and displayed as shown in Figure 6.

The computational performances of CVZ and MPH were also observed during this stage. Each algorithm’s computational performance was observed by recording the refresh rate of the algorithm’s identification component. Specifically, this was achieved by measuring the number of output the models returned per second. The output rate was measured with both one and two hands on screen. MPH maintained a refresh rate of 13.22 outputs per second with two hands in frame and 14.04 with one hand in frame. CVZ maintained an output rate of 14.1 with one hand in frame and, as mentioned, would crash whenever two hands came into frame. 

#### 3.1.2. Qualitative Analysis Results

Out of the three algorithms, two had been successfully implemented, these being MPH and CVZ. The results of the qualitative analysis of both algorithms are detailed below. 

##### Resistance to Translation

Both MPH and CVZ were exposed to a simple translation of a closed fist, facing the camera, around the cameras frame. This involved movements both toward and away from the camera (changing scale) and movements vertically and horizontally across the images frame (traditional translation). Neither algorithm showed signs of major landmark deviation or angular fluctuation during this test. 

##### Resistance to Rotation

Both algorithms were presented with a closed fist with the palm facing the camera, the fist was then rotated about the axis along the operator’s forearm. Throughout this rotation, both MPH and CVZ appeared to maintain a high degree of localisation accuracy, as the displayed landmarks never deviated from their respective joints. Despite this, the joint angles being displayed to the terminal did fluctuate greatly. This suggests that whilst the feature-extraction/landmarking method used by both algorithms is robust against rotation, the modelling method used may not be. This demonstrates that neither algorithm is entirely robust against rotation, which is an issue that may need to be addressed by the classification component. 

##### Resistance to Self-Occlusion

In this test, the hand was held with the palm directly facing the camera, being positioned directly along its axis of reception. Then, the hand was slowly tilted forward until the fingers were directly facing the camera and the palm was completely obscured by the fingers. MPH performed relatively well during this test, as its gesture-tracking component was able to keep an understanding of the hand’s position even after the palm had become completely obscured by the fingers. MPH was also able to maintain sensical joint readings throughout this entire process. CVZ, on the other hand, did not fare as well during this test. Once the hand was approximately 15° off its final position, the bounding box (pink box displayed in Figure 6) raised an exception within the code, crashing the observation algorithm. Thus, the conclusion of this third test was that MPH demonstrated robustness toward self-occlusion and, in its present form, CVZ did not.

#### 3.1.3. Final Selection

From the results detailed in Section 3.1.1 and Section 3.1.2, MPH was selected as the final gesture-identification component. MPH was selected because it was operationally stable, the least computationally intensive, and demonstrated a high degree of robustness when presented with translation and self-occlusion. These results demonstrate the potential that MPH has to aid in the satisfaction of Criteria 1, 2, and 5.

### 3.2. Gesture-Identification Validation Results

#### 3.2.1. Measured Joint Angles

Table 1, Table 2 and Table 3 display the average joint-angle readings recorded for each of the three poses. These joint-angle readings are the averaged results taken from the three participants that were analysed during this study. The tables are organised to display the measured angles for each joint, metacarpophalangeal (J1), proximal interphalangeal (J2), and distal interphalangeal (J3), for each finger.

#### 3.2.2. Calculated Joint Angles

Table 4, Table 5, Table 6, Table 7, Table 8, Table 9, Table 10, Table 11, Table 12, Table 13, Table 14 and Table 15 display the joint angles calculated from MPH. The angles displayed are the average angles that were returned from the three participants that completed testing. The tables are broken up into three groups, with each group containing the angles recorded for that pose. Within each group, there are four tables, with each table containing data calculated from one viewpoint. Each table contains the joint angles calculated using both the two-dimensional and three-dimensional methods of calculations. The tables are organised to display the calculated angles for each joint, metacarpophalangeal (J1), proximal interphalangeal (J2), and distal interphalangeal (J3), for each finger.

#### 3.2.3. Final Accuracy Percentages

As can be seen in Table 16, Table 17, Table 18 and Table 19 below, the data in Section 3.3.1 and Section 3.3.2 were compared and condensed to generate two separate sets of accuracy data. The tables below display the average and minimum accuracy percentages across each viewpoint and across each finger. The tables were further reduced to generate final set of figures for the 3D accuracy, which had an average accuracy of 86.7% and a minimum accuracy of 41.5%, and the 2D accuracy, which had an average accuracy of 83.8% and a minimum accuracy of 9.8%.

### 3.3. Gesture-Classsifier Selection

#### 3.3.1. Scope Definition

As discussed in Section 2.6, there were two scopes of prospective classifiers available for investigation. Given that the results of Stage 4 demonstrated the MPH’s module instability, a second scope of algorithms were selected. In total, four pre-trained classifiers, which utilised different classification techniques and had different computational loads, were selected for further investigation under this section. The four algorithms are listed below: 23.ANN classifier: Conventional machine-learning classifier built to classify Indian sign language—sourced from [32].24.Linear-regression classifier: Conventional machine-learning classifier built to classify Russian sign language—sourced from [33].25.SVM classifier: Conventional machine-learning classifier built to classify Indian sign language—sourced from [34].26.Bounds-based classifier: Non-machine-learning, statically defined classifier built to classify ASLAN counting gestures—altered version, original found from [31].

As per Section 2.6, a common data set was developed such that the four classifiers listed above could be directly compared against one another. After analysing the three gesture sets recognisable by the four classifiers, it was found that they share eight common gestures. A single participant was then imaged to generate the required ten sub-images, which were subsequently modelled using MPH to form the final data set, as shown in Appendix B and Appendix C.

#### 3.3.2. Accuracy Results

Figure 7, Figure 8, Figure 9 and Figure 10 display the confusion matrices used to quantitatively compare the classification accuracies of the four classifiers. The raw data from which these classifiers are generated can be found through Appendix D. From the figures shown below, the non-machine-learning bounds-based classifier performed the best on the given dataset, with an accuracy of 96.25%. The second-best performing classifier was the SVM approach with an accuracy of 81.3%, followed by ANN with an accuracy of 77.5%, and the linear regression model with an accuracy of 70%.

#### 3.3.3. Computational Performance Results

The purpose this section of the results was to quantitatively compare the computational performance of the four potential classifiers. As stated in Section 2.6, this was achieved via a comparison of the time taken by each of the classifier to classify each of the 80 images. Figure 11 displays the computational performance data generated during this testing. As displayed, the bounds-based classifier had the lowest classification time, averaging 81.4 ms. The SVM and linear-regression classifiers had comparable performances, both averaging 88 ms. The slowest of the four algorithms was the ANN model with an average classification time of 196 ms.

#### 3.3.4. Final Classifier Selection

As per the selected scope for this investigation, the selection of the classifier aimed to favour classification accuracy over computational speed. This preference was made to ensure the final algorithm conformed with Governing Criteria 1 and 2. As such, the bounds-based classifier was selected because it had the highest classification accuracy. Fortunately, the bounds-based classifier also had the lowest classification time, indicating that it had the lowest computational requirement, suggesting that it also best conformed to Governing Criterion 5.

### 3.4. Gesture-Mapping Selection and Tuning

With the selection of the gesture classifier complete, the HGR-based HCI component for the alternative-control solution was complete. The final developmental stage of the investigation could begin, with the development of the gesture-mapping component. As defined in Section 2.7, the development of this gesture-mapping component consisted of two main stages, the first being the selection of the command set, and the second stage being the tuning of the mapping component. This investigatory stage began with the implementation of the position command system. 

Following the analysis performed in Stage 5, it was known that the HGR algorithm could accurately recognise the eight gestures present in the common data set. As only eight gestures were required to operate the positional control system, one-to-one mapping could be applied to construct the gesture dictionary shown in Table 20.

Through the implementation of a third-party, open source, python API [35], the above commands were able to be transmitted to the TELLO drone. The positional based control framework was then tested and tuned, using the flight paths defined in Figure 12. 

The most immediate concern from this initial implementation was the overloading of the TELLO’s onboard command buffer. This was caused because the commands were being transmitted at too great a rate for the TELLO drone to process. To solve this, the time taken to recognise a gesture was increased to 500 ms. Whilst this did make the drone appear slightly less responsive initially, it successfully prevented further overloading of the buffer. 

However, after these changes, the TELLO appeared to have periods of unresponsiveness during the testing in the final flight path. These periods of unresponsiveness were accompanied by the TELLO drone continually reporting ‘error No valid IMU’. This error was attributed to the use of the positional command system, and the method the TELLO drone used to process incoming commands. If a command was received by the drone whilst the drone was processing/reading its IMU data, the drone would abandon the task and attempt to execute the command. As positional commands required IMU data to execute, whenever this interrupt situation would occur, the command would fail, and the drone would re-execute its previous command.

To combat this, the gesture dictionary was redefined to use the velocity command system, as shown in Table 21. This command system did not require readings from the IMU sensor; the TELLO drone executed the received commands immediately and consistently. This resulted in the drone appearing more responsive and subsequently easier to control. The same flight paths were used to tune the velocity control system.

The tuning for the velocity control system was the same as the tuning process used for the positional control system. The gestures’ hold time remained the same, whilst the command magnitude and command refresh times were both reduced. After this tuning process, the final alternative-control algorithm was able to easily guide the drone through the defined flight paths. 

### 3.5. Performance Overview

With the completion of Stage 6, the final alternative-control algorithm had been fully developed and proved to be functional. To ascertain the final performance of the solution, it was re-analysed against the governing criteria. 

27.Reliability in issuing the intended command: From the results of Stage 5, the final solution proved to be capable of accurately distinguishing between the intended commands within the command set. When combined with the mapping medium developed in Stage 6, the solution was able to reliably transmit the intended commands to the chosen application medium. Hence, this criterion is satisfied.28.Reproducibility of the intended command: From the results of Stage 4, the initial confidence for the final solution’s conformity to this criterion was challenged due to MPH lack of rotational robustness. However, through the implementation of a bounds-based classifier, the final solution was able to recognise commands robustly and repeatably despite viewpoint and scale variations. Hence, this criterion is satisfied.29.Physically non-restrictive equipment or instrumentation: Given the selection made in Stage 2, the use of a single RGB camera ensured the final solution’s conformity to this criterion. Furthermore, as per the results of Stage 5, the final HGR algorithm is capable of recognising gestures from multiple viewpoints meaning the operator does not have to maintain a perfect position in front of the camera. Hence, this criterion is satisfied.30.Ease of operation and shorter user learning cycle: Given the selection made in Stage 1, the use of single-hand, static, sign language gestures ensured that the commands were simple and easy to learn. When combined with the finely tuned, one-to-one gesture-mapping component developed in Stage 6, natural and accessible drone control was facilitated. Hence, this criterion is satisfied.31.Computationally inexpensive: Through the computational analysis performed in Stage 5, and the selection of MPH justified in Stage 3, the final solution was specifically selected to be as computationally lightweight as possible. Hence, this criterion is satisfied.32.Monetarily inexpensive: Given Stage 2’s selection of an inexpensive, non-specialised data-acquisition method, and the low computational requirements of the final algorithm, the final solution is monetarily efficient. Hence, this criterion is satisfied.

## 4. Discussion

### 4.1. Principal Findings

As previously mentioned, the MPH model accuracy was validated against goniometer readings. The goniometer readings used as a baseline are illustrated in Table 1, Table 2 and Table 3. The joint angles calculated from the MPH model are illustrated in Table 4, Table 5, Table 6, Table 7, Table 8, Table 9, Table 10, Table 11, Table 12, Table 13, Table 14 and Table 15. The comparison of these values produced the model accuracy percentages values summarized in Table 15, Table 16, Table 17 and Table 18. From these accuracy percentages, the Z-axis instability of the MPH model was characterized (explained in greater depth in Section 4.2.2). This characterization was then used to inform the selection of an initial list of prospective classifiers that were theorized to be able to handle this instability. The classification accuracy of these prospective algorithms was then tested using a common input data set, the results of this test are displayed in Figure 7, Figure 8, Figure 9, Figure 10 and Figure 11. The final accuracy performance proved that despite MPH’s drawbacks, it was still applicable in an alternative-control setting as the now-complete HCI functioned at a high confidence level. Section 3.4 then demonstrated how the control-mapping component was created and tuned to apply the fully developed HCI to drone control. The final performance of the algorithm was then judged against its original governing criteria and demonstrated in the attached multi-media video in the Appendix A section. 

### 4.2. Results Analysis

#### 4.2.1. Gesture-Identifier Selection Analysis

These results demonstrate the potential MPH has to aid in the satisfaction of Criteria 1, 2, and 5. As stated in the methodology defined in Section 2.4, the purpose of Stage 3 was to select a gesture-identification algorithm. Stage 3 achieved its purpose through its selection of MPH based on the reasoning detailed in Section 3.1.3. As this was only a threshold-qualitative investigation was simply intended to determine the applicability of the selected solution, it had numerous limitations. The three significant limitations are the lack of a quantitative analysis, the lacklustre computational power of the host machine, and the limited investigation scope. 

The first limitation of this stage was the use of a qualitative analysis method. Whilst this was a valid baseline approach to observe and compare the algorithm’s general localisation accuracy, it fails to provide a solid metric from which the true localisation accuracy of MPH can be extrapolated. The primary constraint this imprints onto the investigation is that there are no quantifiable data proving that MPH is the most accurate solution out of the prospective algorithms. The accuracy of MPH modelling method will be analysed in Stage 4, which does partially account for this. However, the true accuracy of MPH localisation of its landmarks to the joints of the human hand remains largely unknown.

The second limitation arises from the lack of computational power provided by the host machine. This limitation directly excluded InterHands2.6M, which has been proven to be a feature-extraction method comparable to MPH [19]. Whilst MPH did successfully conform to the governing criteria relevant to this section, InterHands2.6M may have had a superior accuracy or computational performance once trained. Unfortunately, this limitation was unavoidable as the host machine chosen for this paper was the only machine available.

The third limitation is derived from the limited number of independent variables that the tested gesture-identification algorithms were tested against. In an ideal setting, if Stage 3 was to be expanded, the gesture-identification algorithms should also have been tested against other sources of variance that are likely to be included in alternative-control applications. These include but are not limited to background variation, operator hand colour variation, operator hand size variation, and lighting variations. Whilst the failure to include these does not invalidate the selection of MPH, it should be considered for future work.

#### 4.2.2. Gesture-Identifier Validation Analysis

As stated in the methodology section defined in Section 2.5, the primary purpose of this stage was to validate the performance of the gesture-identification algorithm. The results shown in Stage 4 achieved this purpose through the successful application of a clinical method to ascertain the accuracy of the MPH model. The results of this investigation effectively displayed the advantages and disadvantages of MPH, the disadvantages of which must be mitigated by future components to ensure the final solution’s conformity to the governing criteria. 

From the data recorded during Stage 4, four trends about MPH model accuracy became immediately apparent: confirmation of the high potential accuracy of MPH modelling systems, the model’s susceptibility to rotation and pose due to its normalised depth coordinate system, the innate benefit of the three-dimensional modelling system over a conventional two-dimensional system, and the model’s resistance to self-occlusion. Each of these observations had considerable implications for the future development of both this alternative-control algorithm and for future applications seeking to implement MPH (future implementations will be discussed in greater detail in Section 5).

The first key finding from the analysis performed during Stage 4 was the positive observation of MPH potential accuracy. This key finding supported the observations made in Stage 3, which are demonstrated in Table 16 and Table 17. From these tables, MPH was shown to be capable of accurately observing the digits of the human hand to a high degree, having a maximum accuracy percentage of 96.2% while maintaining an average observational accuracy of 86.7%. This alludes to MPH’s capability to satisfy Criterion 1, if coupled with an appropriate classification algorithm that can handle the major disadvantages of MPH which will be discussed next. 

The second key finding was the observation that MPH robustness was severely limited by its “2.5D coordinate system”. MPH uses a normalised depth component, meaning that while the x and y components of a landmark’s coordinate are derived from the image’s width and height, the z component is derived from a depth calculation between the landmark and the wrist of the observed hand. This results in a model that is not a true three-dimensional representation of the observed hand because the z coordinates have a different scale to the x and y coordinates. MPH attempts to adjust for this by normalising the z component to be within a similar numerical range to the x and y components. However, as demonstrated in Table 18 and Table 19, the percentage accuracy of the model can be seen to vary greatly based upon the importance of the z coordinate in the angle calculations. 

Orientations that placed connected coordinates along the x and y planes with limited changes in the z coordinate had notably higher accuracy values than orientations that had large changes in the z coordinate. In other words, if the two vectors from which an angle is calculated had large differences in their respective z values, i.e., the angle being observed lay upon the z, y plane, the resultant angle would have a low accuracy. A good example of this observation is a comparison between the percentage accuracy of the closed pose when observed from the side and from the front. When observed from the front, the average accuracy of the MPH model was only 67.8%, in comparison to the model’s 84.8% accuracy when observed from the side. From this demonstration, it can be inferred that the normalisation method implemented by MPH fails to successfully equalise the relative scales of the x, y, and z planes, causing the aforementioned instability to rotation. This observation raises a concerning disadvantage for MPH. If this disadvantage is not mitigated by future components, it will have to be mitigated by the operator. Specifically, to ensure that Governing Criterion 1 is met, the operator will have to keep their hands in a constant orientation with respect to the camera. This both limits the manoeuvrability of the operator and makes the algorithm more difficult to use, jeopardising the final algorithm’s conformity to Governing Criteria 3 and 4. 

The third key finding was that despite the drawbacks of the 2.5D modelling system, the depth component did have some notable benefits over a traditional 2D model. This observation came from the direct comparison between the 2D angle calculations which ignored the depth component and the 3D calculations which used the depth component. While both systems struggled to handle rotation, the three-dimensional system was notably more stable than the two-dimensional system. This can be directly observed by the overall percentages stated in Section 3.2.3, whereby the three-dimensional system’s minimum observed accuracy only dropped to 41.5% whilst the two-dimensional system fell to 9.8%. This key finding was practically relevant when informing the selection of future components. Future components were selected to avoid reductions in MPH model complexity, such as the use of joint angles directly, or the removal of the z coordinate, instead favouring the use of the full 21-landmarks, x, y, z coordinate system. 

The final key finding was the quantitative validation of MPH resistance to self-occlusion. This key finding validated MPH’s ability to handle one of the key limitations of single camera RGB data acquisition, reinforcing its selection as the final gesture-identification algorithm. Evidence for this key finding can be observed in Table 18 and Table 19, whereby the average and minimum joint accuracies calculated from the back and side viewpoints remain comparable, if not favourable to angles calculated from the front on viewpoints. As can be seen in Appendix B and Appendix C, many of the images taken from the side and back viewpoint had fingers that were not able to be directly observed because they were obscured by other parts of the hand. The fact that the angles calculated from these images produced results similar, if not superior, to their non-obscured counter parts proves MPH’s robustness against self-occlusion. This key finding again supports MPH’s selection, as it demonstrates how it conforms to Governing Criteria 2 and 4. 

However, during the Stage 4 investigation, two limitations were encountered. These were the limited fidelity of the goniometer measurements and the limited number of poses used. The goniometer used for this experiment had a measuring fidelity of 5° intervals. These intervals meant that in almost every measurement, the joint angle was being rounded to the nearest 5° mark as the observed angle often fell between these said marks. This rounding could have contributed to the error percentages recorded in Stage 4. However, as non-digital goniometers remain the industry standard for joint measurements [36], this source of error had to be accepted. This leads into the second limitation that impacted the Stage 4 investigation. Due to the specification to only use three governing poses to keep consistency among all participants, these poses could not be adjusted to set the goniometer to its nearest 5° mark. This prevented the immediate resolution of the error source above. If a more flexible posing regime had been implemented, the poses could have been altered on a participant-by-participant basis to ensure that the goniometer was reading whole values, rather than having to round to the nearest value. 

#### 4.2.3. Gesture-Classifier Selection Analysis

As stated in the methodology for Stage 5, the primary purpose of this stage was to select a gesture classifier that best complimented the functionality of the selected gesture-identification algorithm. The results shown in Stage 5 achieved this purpose through the selection of a bounds-based classifier based on the reasoning shown in Section 3.3.4. Aside from achieving its key purpose, this section produced another key finding which supported the selection of MPH as the gesture-identification component. However, this investigation stage also had three key limitations: the impact of gesture set reduction on pre-trained classifiers, the tests’ limited data set, and the unexplained variations in the computational performance data.

The most important finding produced by this investigatory stage, aside from the selection of the classification algorithm, was the demonstration that MPH’s disadvantages are surmountable. As can be seen from the confusion matrices shown in Figure 7, despite the inherent variations in the MPH model, the bounds-based classifier was able to achieve an extremely high level of accuracy. 

This shows that when MPH is combined with the appropriate auxiliary systems, its disadvantages can be minimised, maximising its effectiveness as a gesture-identification algorithm. This was a key finding that served to both validate the selection of MPH within this overall solution and look favourably on the implementation of MPH in future solutions.

The first key limitation that Stage 5 faced was the impact of gesture set reduction on a pre-trained classifier. As the classifiers could not be re-trained to recognise the eight newly defined gestures, a set of external controls had to be implemented to reduce their classification ranges. This reduction came in the form of a simple cascading set of ‘if–else’ statements that forced the classifiers to return the highest probability gesture out of the eight-gesture subset, even if the classifier would have otherwise returned a different gesture. This reduction had to be made to produce a proper confusion matrix. To prevent this reduction from impacting upon the overall validity of this experiment, the time take to complete this reduction was removed from the overall classification time. Ideally, the classifiers would have been re-trained to only recognise the eight new defined gestures. 

The second key limitation was the size and scope of the data set. Given that this was one of the two secondary investigations performed over the course of this paper, it was not deemed practical to develop a test data set with more than a thousand images. This limited the level of testing that could be performed on the four classifiers. While it does not invalidate the classifier selection made above, in an ideal setting, a larger data set would have been used, with more edge-case gestures being included to really test the limits of the classification algorithms. 

The final limitation centred around the unexpected behaviour displayed toward the end of Figure 11. Initially, it was assumed that each of the algorithm’s classification times would remain largely consistent across the 80 input images. This was an assumption based on the idea that the classification time would only fluctuate greatly if the size and/or complexity of the input data changed. As the input data remained constant, in both size and complexity across the 80 input models, the fluctuation remained unexplained. However, as the selected classifier was consistently faster than the other classifiers even in the regions of fluctuation, the validity of the final solution was not compromised by this limitation.

## 5. Conclusions

This project achieved its primary purpose by developing a functioning alternative-control algorithm that extended the usability of a quad-rotor drone. This was achieved through the development of an HGR algorithm that combined the functionality of MPH and a bounds-based classifier. The final solution facilitated natural and accessible control while being computationally inexpensive and not requiring the use of specialized camera equipment. The success of the final solution demonstrated the applicability of modern, single-camera HGR algorithms within the confines of alternative control. Furthermore, the clinical evaluation of MPH demonstrated MPH’s inherent advantages and disadvantages. The success of the developed alternative-control algorithm shows that when handled appropriately, MPH can be a powerful HGR tool that has applications within clinical and control settings. However, future projects seeking to apply MPH must be mindful of the algorithm’s limitations or risk failure. There are three main areas of future work related to this paper, these being an extension of the applied methodology, the application of MPH’s model validation data, and the application of the developmental framework used in this paper alongside the final solution itself. 

The first proposed area of future work is an extension of the method applied in this paper. One area of this extension is the completion of a broader comparative review of gesture-identification components, specifically aiming to extend upon the works of this paper by including a quantitative comparison of landmark-localization accuracy in its analysis of algorithms such as MPH and InterHands2.6M. Furthermore, another proposed extension of this paper’s methodology is a data-based analysis of the final solution’s performance. This could be achieved by grading the final solution and the TELLO drone’s standard control mechanism against the governing criteria of this paper using quantitative metric-based testing. 

The second proposed area of future work is the application of the MPH model’s validation data and joint-localization accuracy observations. The first area of application is in clinical diagnoses, focusing on using MPH to generate joint angles accurate enough to diagnoses illness and injury. As per the findings of this paper, MPH cannot do this directly due to its modelling method’s instability; however, it could be achieved using sensor fusion. One possible avenue would be using MPH’s joint-localisation accuracy to efficiently locate points of interest and then modelling these joints in a true three-dimensional environment using, for example, a stereo camera. The second setting is in rehabilitation, focusing on using MPH’s current level of accuracy to observe the general motion of the human hand and then acting upon this motion in a gamified environment.

The final proposed area of future work is the application of the developed alternative-control algorithm and the framework used to construct it. The most immediate application for the developed alternative-control algorithm is using it to extend the accessibility of drone control to operators that cannot operate the standard control medium due to having impaired dexterity. Another application for the developed algorithm is seeking to optimise its computationally lightweight nature to foster its use in either embedded or edge computing environments. The framework used to develop the final alternative-control algorithm can be re-applied to a multitude of other alternative-control tasks. With the now fully developed HGR component, all that would be required to re-tune the current solution to control a new agent is the reconstruction of the command mapping component.

## Figures and Tables

**Figure 1 sensors-23-05462-f001:**
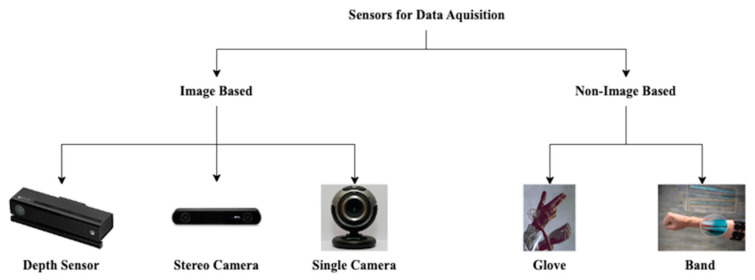
HGR data-acquisition categories.

**Figure 2 sensors-23-05462-f002:**
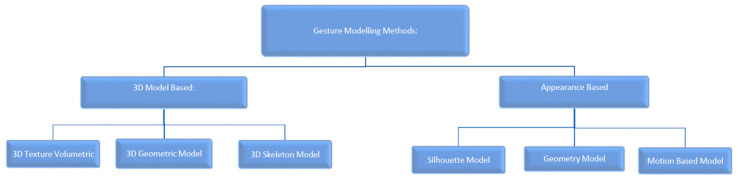
Hand-gesture modelling methods.

**Figure 3 sensors-23-05462-f003:**
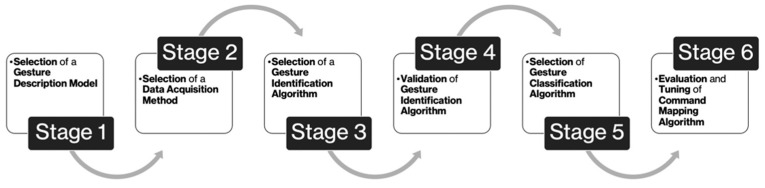
Investigation structure flow chart.

**Figure 4 sensors-23-05462-f004:**
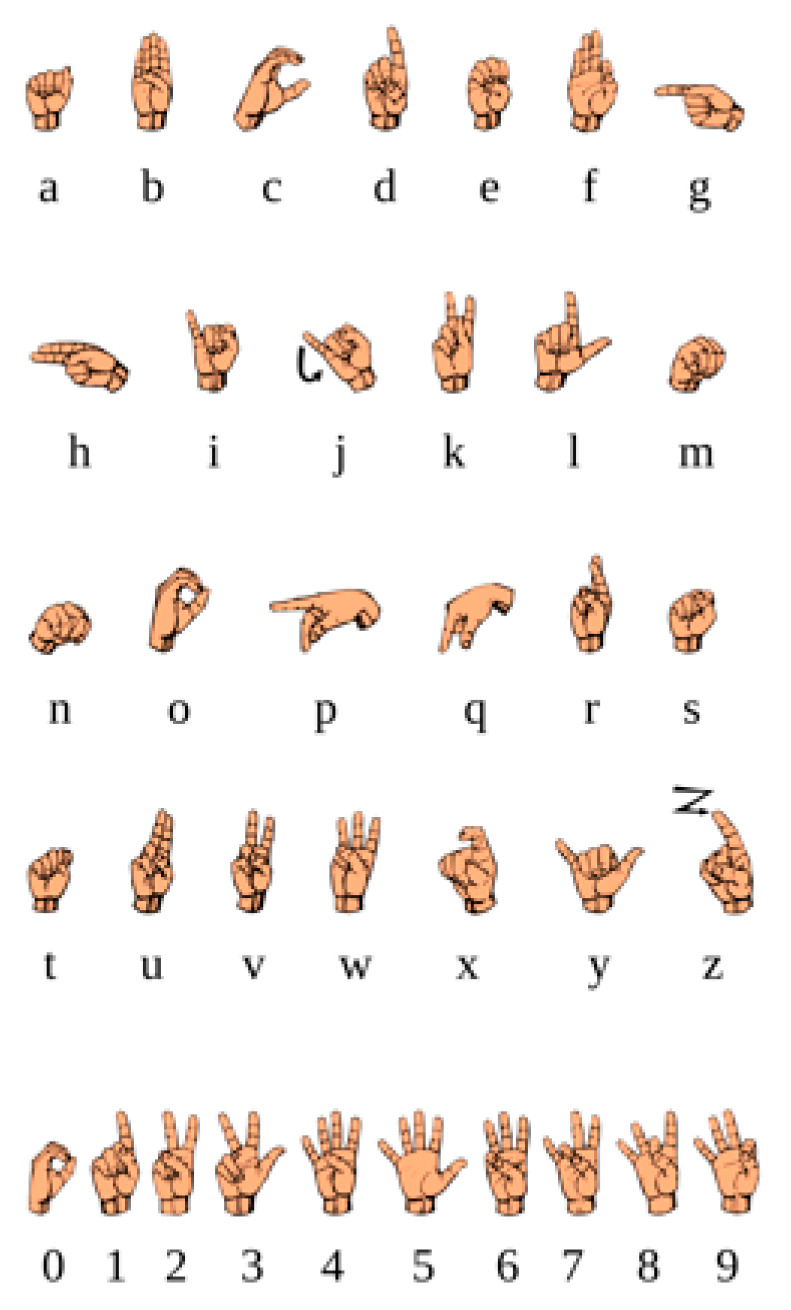
Sign language example hand-gesture set [23].

**Figure 5 sensors-23-05462-f005:**
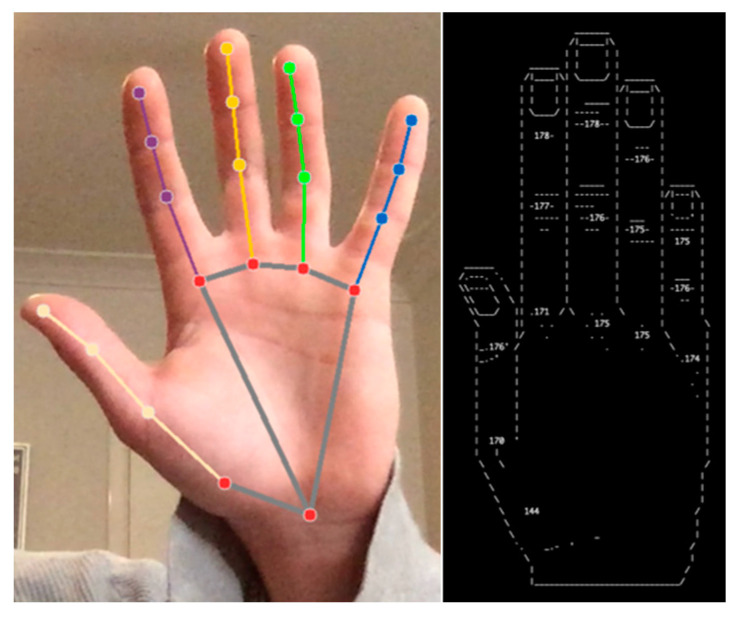
MPH output displaying the 21 coloured landmarks overlayed onto the input image and the text based joint angle measurement model displayed through terminal.

**Figure 6 sensors-23-05462-f006:**
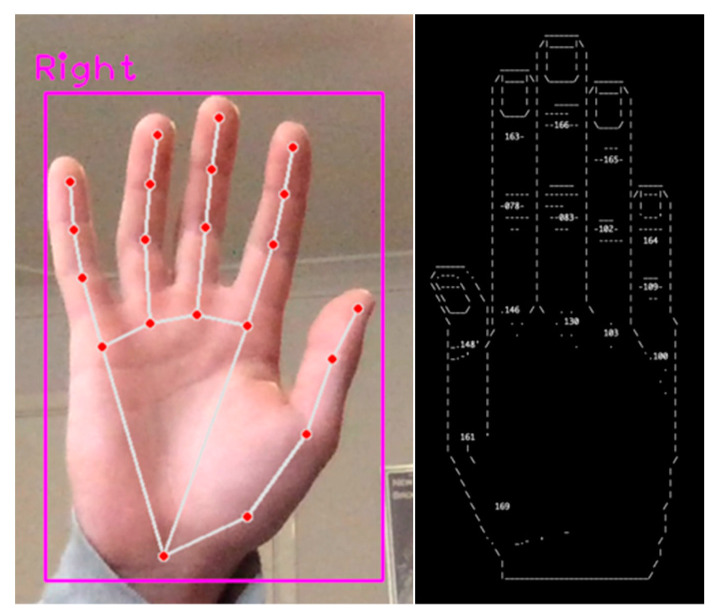
CVZ output displaying the 21 red landmarks overlayed onto the input image and the text based joint angle measurement model displayed through terminal.

**Figure 7 sensors-23-05462-f007:**
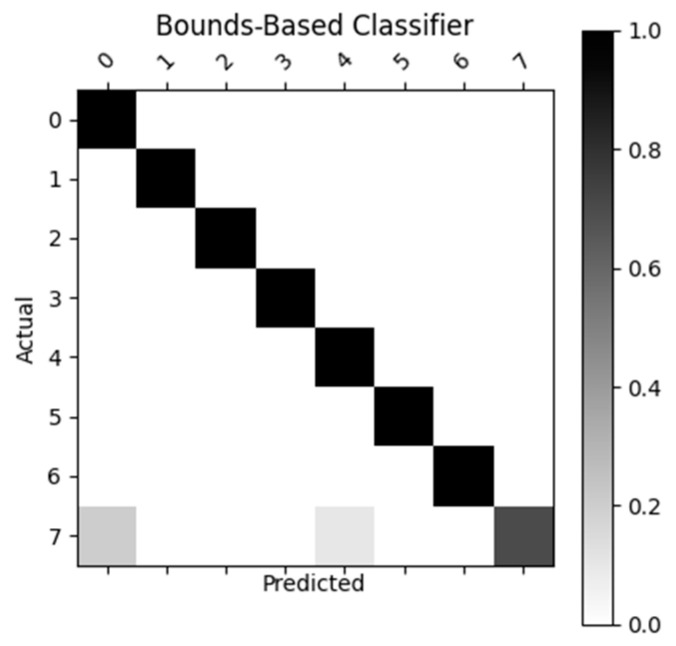
Confusion Matrix 1—bounds-based classifier accuracy.

**Figure 8 sensors-23-05462-f008:**
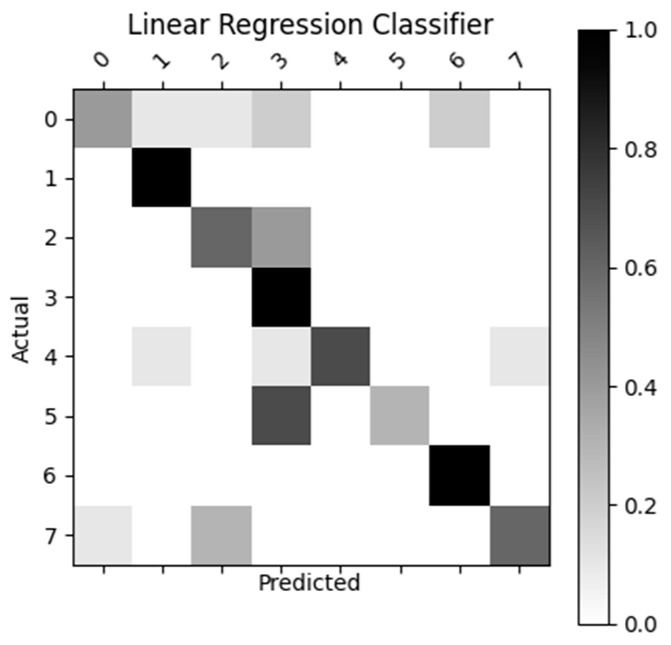
Confusion Matrix 2—linear-regression classifier accuracy.

**Figure 9 sensors-23-05462-f009:**
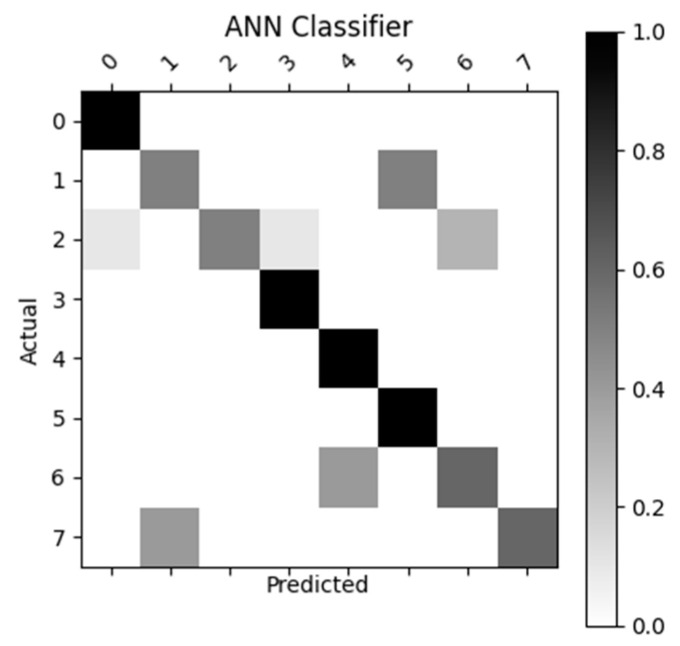
Confusion Matrix 3—ANN classifier accuracy.

**Figure 10 sensors-23-05462-f010:**
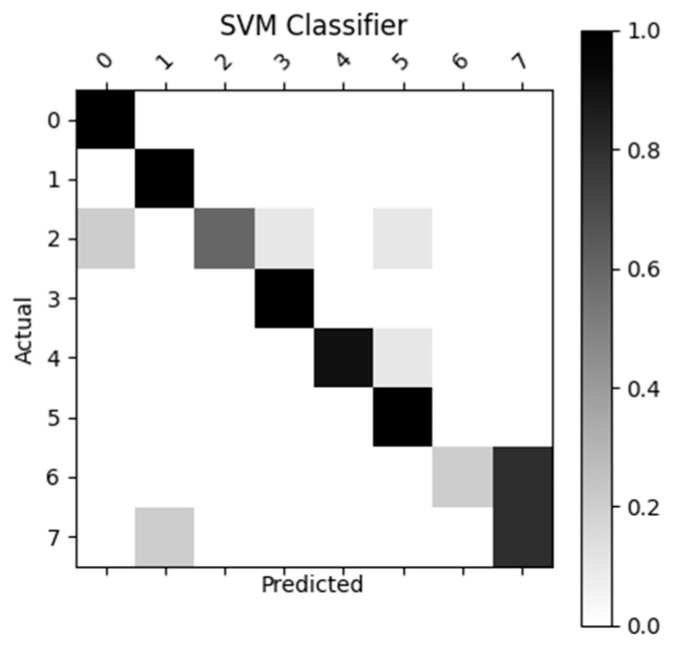
Confusion Matrix 4—SVM classifier accuracy.

**Figure 11 sensors-23-05462-f011:**
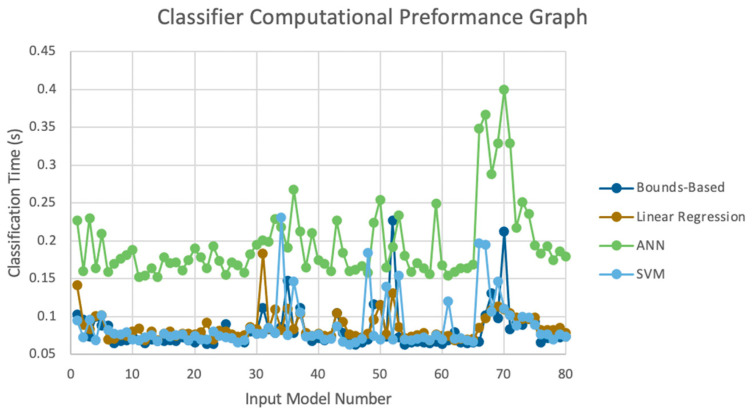
Comparison of tested classifiers computational performance.

**Figure 12 sensors-23-05462-f012:**
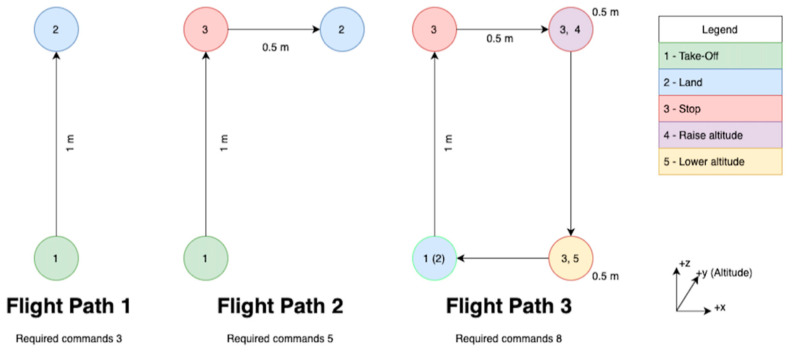
Test flight paths.

**Table 1 sensors-23-05462-t001:** Open-palm joint angles measured by goniometer.

Joint Number	Thumb	Index	Middle	Ring	Pinkie
J1	162	178	175	178	178
J2	177	172	172	170	172
J3	180	177	180	180	180

**Table 2 sensors-23-05462-t002:** Partially closed palm joint angles measured by goniometer.

Joint Number	Thumb	Index	Middle	Ring	Pinkie
J1	153	168	168	170	173
J2	157	93	92	87	97
J3	118	113	107	117	113

**Table 3 sensors-23-05462-t003:** Closed-palm joint angles measured by goniometer.

Joint Number	Thumb	Index	Middle	Ring	Pinkie
J1	148	98	98	107	107
J2	143	82	83	82	90
J3	117	110	108	108	112

**Table 4 sensors-23-05462-t004:** Closed-palm joint angles calculated from the MPH model—front.

Joint Number	Thumb		Index		Middle		Ring		Pinkie	
	3D	2D	3D	2D	3D	2D	3D	2D	3D	2D
J1	151	156	142	142	144	144	142	140	132	128
J2	151	151	53	54	47	48	34	35	40	41
J3	105	100	164	161	127	115	136	128	147	143

**Table 5 sensors-23-05462-t005:** Closed-palm joint angles calculated from the MPH model—45°.

Joint Number	Thumb		Index		Middle		Ring		Pinkie	
	3D	2D	3D	2D	3D	2D	3D	2D	3D	2D
J1	140	138	133	157	123	133	112	112	91	85
J2	144	150	78	63	73	63	66	67	83	89
J3	147	148	133	132	120	123	120	120	118	111

**Table 6 sensors-23-05462-t006:** Closed-palm joint angles calculated from the MPH model—side.

Joint Number	Thumb		Index		Middle		Ring		Pinkie	
	3D	2D	3D	2D	3D	2D	3D	2D	3D	2D
J1	143	149	74	76	73	73	76	75	83	82
J2	154	163	107	103	95	94	92	93	95	95
J3	163	163	113	108	113	113	114	113	113	110

**Table 7 sensors-23-05462-t007:** Closed-palm joint angles calculated from the MPH model—back.

Joint Number	Thumb		Index		Middle		Ring		Pinkie	
	3D	2D	3D	2D	3D	2D	3D	2D	3D	2D
J1	159	161	125	135	121	156	116	148	108	118
J2	154	154	75	25	90	20	97	37	94	54
J3	105	105	163	170	155	173	143	170	157	163

**Table 8 sensors-23-05462-t008:** Partially closed palm joint angles calculated from the MPH model—front.

Joint Number	Thumb		Index		Middle		Ring		Pinkie	
	3D	2D	3D	2D	3D	2D	3D	2D	3D	2D
J1	152	154	145	156	154	164	161	176	156	160
J2	161	161	119	97	99	60	103	75	110	94
J3	108	105	124	98	138	132	126	116	122	115

**Table 9 sensors-23-05462-t009:** Partially closed palm joint angles calculated from the MPH model—45°.

Joint Number	Thumb		Index		Middle		Ring		Pinkie	
	3D	2D	3D	2D	3D	2D	3D	2D	3D	2D
J1	141	142	145	166	158	171	165	170	151	151
J2	164	167	100	87	85	76	78	74	92	88
J3	132	132	129	126	127	124	127	128	134	135

**Table 10 sensors-23-05462-t010:** Partially closed palm joint angles calculated from the MPH model—side.

Joint Number	Thumb		Index		Middle		Ring		Pinkie	
	3D	2D	3D	2D	3D	2D	3D	2D	3D	2D
J1	145	134	111	121	113	114	116	115	110	115
J2	142	146	113	113	107	106	104	104	121	122
J3	161	160	139	132	133	132	135	134	145	146

**Table 11 sensors-23-05462-t011:** Partially closed palm joint angles calculated from the MPH model—back.

Joint Number	Thumb		Index		Middle		Ring		Pinkie	
	3D	2D	3D	2D	3D	2D	3D	2D	3D	2D
J1	160	163	151	159	162	169	153	159	144	150
J2	153	153	100	89	84	38	91	63	102	85
J3	118	116	151	98	156	153	154	135	146	121

**Table 12 sensors-23-05462-t012:** Open-palm joint angles calculated from the MPH model—front.

Joint Number	Thumb		Index		Middle		Ring		Pinkie	
	3D	2D	3D	2D	3D	2D	3D	2D	3D	2D
J1	159	165	163	166	168	172	171	179	166	170
J2	175	175	168	175	169	178	171	173	171	171
J3	166	166	177	178	178	179	174	177	172	174

**Table 13 sensors-23-05462-t013:** Open palm joint angles calculated from the MPH model—45°.

Joint Number	Thumb		Index		Middle		Ring		Pinkie	
	3D	2D	3D	2D	3D	2D	3D	2D	3D	2D
J1	149	149	151	167	164	171	171	172	168	169
J2	171	176	165	173	163	171	163	164	157	157
J3	169	170	175	175	175	175	175	175	172	172

**Table 14 sensors-23-05462-t014:** Open palm joint angles calculated from the MPH model—side.

Joint Number	Thumb		Index		Middle		Ring		Pinkie	
	3D	2D	3D	2D	3D	2D	3D	2D	3D	2D
J1	163	159	140	164	147	168	163	171	161	168
J2	169	174	165	169	157	169	161	171	169	172
J3	174	168	167	177	176	178	172	174	170	172

**Table 15 sensors-23-05462-t015:** Open palm joint angles calculated from the MPH model—back.

Joint Number	Thumb		Index		Middle		Ring		Pinkie	
	3D	2D	3D	2D	3D	2D	3D	2D	3D	2D
J1	159	164	161	161	172	177	162	164	152	154
J2	176	177	174	176	171	177	175	179	168	175
J3	168	169	174	177	177	178	175	176	175	176

**Table 16 sensors-23-05462-t016:** Model accuracy calculated from 3D data—by finger.

Hand Position	Percentage Accuracy									
	Thumb		Index		Middle		Ring		Pinkie	
	Avg	Min	Avg	Min	Avg	Min	Avg	Min	Avg	Min
Open	96.2%	92.0%	93.8%	78.7%	95.7%	84.0%	95.7%	91.0%	94.4%	85.4%
Partial	92.2%	63.6%	82.0%	66.1%	82.6%	54.2%	86.2%	68.2%	82.6%	63.6%
Closed	89.6%	60.7%	72.3%	50.9%	77.0%	53.1%	78.5%	41.5%	82.3%	44.4%

**Table 17 sensors-23-05462-t017:** Model accuracy calculated from 2D data—by finger.

Hand Position	Percentage Accuracy									
	Thumb		Index		Middle		Ring		Pinkie	
	Avg	Min	Avg	Min	Avg	Min	Avg	Min	Avg	Min
Open	96.2%	92.0%	95.6%	84.3%	96.8%	89.1%	96.2%	92.1%	95.4%	86.5%
Partial	91.2%	64.4%	88.9%	72.0%	77.6%	41.3%	86.8%	67.6%	85.4%	66.5%
Closed	88.5%	60.7%	57%	9.8%	59.7%	16.9%	66.4%	22.0%	76.6%	38.9%

**Table 18 sensors-23-05462-t018:** Model accuracy calculated from 3D data—by viewpoint.

Hand Position	Percentage Accuracy							
	Front		Forty-Five		Side		Back	
	Avg	Min	Avg	Min	Avg	Min	Avg	Min
Open	96.8%	91.6%	94.5%	84.8%	93.2%	78.7%	96.0%	85.4%
Partial	88.6%	71.0%	90.0%	81.3%	76.1%	63.6%	85.7%	54.2%
Closed	67.8%	41.5%	86.3%	64.3%	84.8%	60.7%	80.7%	51.8%

**Table 19 sensors-23-05462-t019:** Model accuracy calculated from 2D data—by viewpoint.

Hand Position	Percentage Accuracy							
	Front		Forty-Five		Side		Back	
	Avg	Min	Avg	Min	Avg	Min	Avg	Min
Open	97.6%	92.2%	96.3%	91.2%	94.1%	84.3%	96.2%	86.5%
Partial	91.3%	65.2%	90.3%	80.5%	76.9%	64.4%	85.6%	41.3%
Closed	53.2%	9.8%	81.9%	39.8%	84.9%	60.7%	58.0%	24.1%

**Table 20 sensors-23-05462-t020:** Positional command set.

Gesture Identifier	Command
1	Move along z axis (forward velocity)
2	Move along −z axis (backward velocity)
3	Move along y axis (upward velocity)
4	Move along −y axis (downward velocity)
5	Move along x axis (bank right)
6	Move along −x axis (bank left)
7	Set required movement along, x, y, and z axes to zero (stop)
8	Take-off or land (depending on whether in flight, or landed)

**Table 21 sensors-23-05462-t021:** Velocity command set.

Gesture Identifier	Command
1	Increase velocity along z axis (forward velocity)
2	Increase velocity along −z axis (backward velocity)
3	Increase velocity along y axis (upward velocity)
4	Increase velocity along −y axis (downward velocity)
5	Increase velocity along x axis (bank right)
6	Increase velocity along −x axis (bank left)
7	Set velocity along, x, y, and z axes to zero (stop)
8	Take-off or land (depending on whether in flight or landed)

## Data Availability

Not applicable.

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
