# Peer review of "Design and Evaluation of an Alternative Control for a Quad-Rotor Drone Using Hand-Gesture Recognition"

_sensors, 2023, doi:10.3390/s23125462_

Round 1
Reviewer 1 Report
The following have to be addressed prior to consideration for publication in Sensors:
1. The abstract does not clearly indicate the contributions and major findings of the study. The first paragraph is an extensive introduction, while the second paragraph provides an overview of the contributions and findings. A more detailed discussion of the contributions and findings is required, and the introductory paragraph can be shorter.
2. The captions of most tables and figures need to be revised and rewritten. The captions should include a comprehensive description of the tables and figures that can be read independently. Additionally, table captions are typically placed above the tables.
3. The reviewer is unsure what is meant by "validated" in line 141: "The final alternative control algorithm was applied and validated using a quad-rotor drone." It is not clear in which part of the paper and how the control algorithm was validated quantitatively using a quad-rotor drone.
4. The sentence "The purpose of this project was to construct" in line 155 is not suitable for a paper and should be changed.
5. Section 2.1.3, "Defining Simplifications," is difficult to follow for readers and should be explained more clearly, especially by using examples of the simplifications.
6. The criteria in section 2.1.4, "Governing Criteria," must be defined by equations or mathematics if they are quantitative. For example, which metrics are used to calculate the second criterion, repeatability, and which computational method is employed?
7. Sections 2.2.2, "Gesture Type Selection Justification," 2.2.3, "Gesture Model Selection Justification," 2.2.4, "Gesture Information Justification," and 2.3, "Stage Two: Selection of Data Acquisition Method," need revision. Each component should be evaluated using a table that displays the values of different criteria for different methods, decisions, or considerations, including the selected option and the ones that are not selected. Selections in these sections are mostly based on qualitative arguments that are not objective, especially as most of the arguments lack a reference.
8. The reason why "single-hand, static gestures will increase the robustness of the final solution" is unclear from the previous lines in line 275.
9. Several abbreviations, such as LHS and RHS, are not well defined and should be corrected.
10. The introduction should be updated with more recent studies and provide a comparison with similar efforts.
See the notes above about the correct choice on captions, and in general the selection of trivial expressions such as for example the last part of the abstract which it would be better saying what are the research methods that scientifically support the work accomplished.
The technical importance of this paper stems from the results produced by the clinically sound evaluation of the gesture identification component used by the final solution, MPH. These results highlighted the key advantages and disadvantages of MPH, the understanding of which was pivotal in the development of the final algorithm and can also be used to inform future projects on the applicability of MPH.
Author Response
Dear reviewer please find Attached below , thank you

Reviewer 2 Report
The following recommendations are given for improving the research work:
1) It is recommended to consider the following materials are listed in references for their relevance to the research work:
- 12. Machine Learning in Radiation Oncology; El Naqa, I.; Li, R.; Murphy, M. (Eds.) Springer: Cham, Switzerland, 2015. doi: 1240 10.1007/978-3-319-18305-3. 1241
- 13. Nath, V.; Mandal, J. K. Nanoelectronics, Circuits and Communication Systems Proceeding of NCCS 2018. In Conference pro- 1242 ceedings NCCS (pp. 31-36). doi: 10.1007/978-981-15-2854-5.
2) The following appendixes can be removed from the research for decrease the volume of the paper: Appendix C, Appendix D
3) Section 2.1.2 of the paper is not given, after 2.1.1 is followed 2.1.3. There are has some mistakes
4) The results of the research work have not been sufficiently compared and evaluated with other methods
5) In the title of the research paper, it is written that the research results will be applied to the control of a Quad Rotor Drone, but this mechanism is not sufficiently written in the article
Minor editing of English language required
Author Response
Attached below

Reviewer 3 Report
Dear Authors
The manuscript raised an important and hot topic in Design and Evaluation of an Alternative Control for a Quad-Rotor Drone using Hand Gesture Recognition. I believe that it is one of the interesting topics. However, I have several doubts concerning this manuscript which in details are given in the attached PDF file.
Regards

Although English is not my native language either, it is notable that some verb conjugations and general terms need revision by the authors.
Author Response
Attached below
